# 🍑 PEACH: PRETRAINED-EMBEDDING EXPLANATION ACROSS CONTEXTUAL AND HIERARCHICAL STRUCTURE

## ABSTRACT

In this work, we propose a novel tree-based explanation technique, PEACH (Pretrained-embedding Explanation Across Contextual and Hierarchical Structure), that can explain how text-based documents are classified by using any pretrained contextual embeddings in a tree-based human-interpretable manner. Note that PEACH can adopt any contextual embeddings of the PLMs as a training input for the decision tree. Using the proposed PEACH, we perform a comprehensive analysis of several contextual embeddings on nine different NLP text classification benchmarks. This analysis demonstrates the flexibility of the model by applying several PLM contextual embeddings, its attribute selections, scaling, and clustering methods. Furthermore, we show the utility of explanations by visualising the feature selection and important trend of text classification via human-interpretable word-cloud-based trees, which clearly identify model mistakes and assist in dataset debugging. Besides interpretability, PEACH outperforms or is similar to those from pretrained models[1].

## 1 INTRODUCTION

Large Pretrained Language Models (PLMs), like BERT, RoBERTa, or GPT, have made significant contributions to the advancement of the Natural Language Processing (NLP) field. Those offer pretrained continuous representations and context models, typically acquired through learning from co-occurrence statistics on unlabelled data, and enhance the generalisation capabilities of downstream models across various NLP domains. PLMs successfully created contextualised word representations and are considered word vectors sensitive to the context in which they appear. Numerous versions of PLMs have been introduced and made easily accessible to the public, enabling the widespread utilisation of contextual embeddings in diverse NLP tasks.

However, the aspect of human interpretation has been rather overlooked in the field. Instead of understanding how PLMs are trained within specific domains, the decision to employ PLMs for NLP tasks is often solely based on their state-of-the-art performance. This raises a vital concern: *Although PLMs demonstrate state-of-the-art performance, it is difficult to fully trust their predictions if humans cannot interpret how well they understand the context and make predictions.* To address those concerns, various interpretable and explainable AI techniques have been proposed in the field of NLP, including feature attribution-based (Ribeiro et al., 2016; Sha et al., 2021; Ribeiro et al., 2018; Luo et al., 2018; He et al., 2019), language explanation-based (Ling et al., 2017; Ehsan et al., 2018) and probing-based methods (Sorodoc et al., 2020; Prasad & Jyothi, 2020; Klafka & Ettinger, 2020). Among them, feature attribution based on attention scores has been a predominant method for developing inherently interpretable PLMs. Such methods interpret model decisions locally by explaining the prediction as a function of the relevance of features (words) in input samples. However, these interpretations have two main limitations: it is challenging to trust the attended word or phrase as the sole responsible factor for a prediction (Serrano & Smith, 2019; Pruthi et al., 2020), and the interpretations are often limited to the input feature space, requiring additional methods for providing a global explanation (Han et al., 2020; Rajagopal et al., 2021). Those limitations of interpretability are ongoing scientific disputes in any research fields that apply PLMs, such as

---

[1]Code and Implementation details will be provided via GitHub after the acceptance.

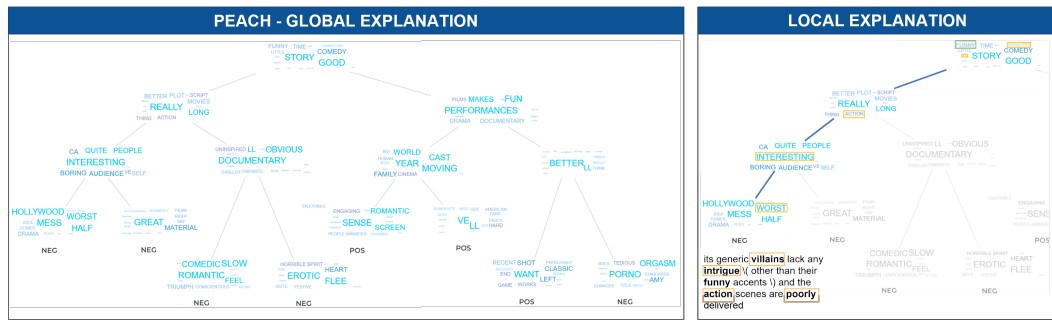

Figure 1: A PEACH is a globally interpretable model that faithfully explains the pretrained models' reasoning using the pretained contextual embedding (left, on MR dataset). Additionally, the decision-making process for a single prediction can be detailed presented (right, partially shown). The detailed description can be found in Section 5.

Computer Vision (CV). However, the CV field has relatively advanced PLM interpretability strategies since it is easier to indicate or highlight the specific part of the image. While several post-hoc methods (Zhou et al., 2018a; Fong & Vedaldi, 2017; Bach et al., 2015; Yeh et al., 2018) give an intuition about the black-box model, decision tree-based interpretable models such as prototype tree (Nauta et al., 2021) have been capable of simulating context understanding, faithfully displaying the decision-making process in image classification, and transparently organising decision rules in a hierarchical structure. However, the performance of these decision tree-based interpretations with neural networks is far from competitive compared to state-of-the-art PLMs (Devlin et al., 2019; Liu et al., 2020).

For NLP, a completely different question arises when we attempt to apply this decision tree interpretation method: *What should be considered a node of the decision tree in NLP tasks?* The Computer Vision tasks typically use the specific image segment and indicate the representative patterns as a node of the decision tree. However, in NLP, it is too risky to use a single text segment (word/phrase) as a representative of decision rules due to semantic ambiguity. For example, if we have a single word 'party' as a representative node of the global interpretation decision tree, its classification into labels like *'politics'*, *'sports'*, *'business'* and *'entertainment'* would be highly ambiguous.

In this work, we propose a novel decision tree-based interpretation technique, PEACH (Pretrained-embedding Explanation Across Contextual and Hierarchical Structure), that aims to explain how text-based documents are classified using any pretrained contextual embeddings in a tree-based human-interpretable manner. We first finetune PLMs for the input feature construction and adopt the feature processing and grouping. Those grouped features are integrated with the decision tree algorithms to simulate the decision-making process and decision rules, by visualising the hierarchical and representative textual patterns. In this paper, our main contributions are as follows:

- Introducing PEACH, the first model that can explain the text classification of any pretrained models in a human-understandable way that incorporates both global and local interpretation.

- PEACH can simulate the context understanding, show the text classification decision-making process and transparently arrange hierarchical-structured decision rules.

- Conducting comprehensive evaluation to present the preservability of the model prediction behaviour, which is perceived as trustworthy and understandable by human judges compared to widely-used benchmarks.

## 2 PEACH

The primary objective of PEACH (Pretrained-embedding Explainable model Across Contextual and Hierarchical Structure) is to identify a contextual and hierarchical interpretation model that elucidates text classifications using any pretrained contextual embeddings. To accomplish this, we outline

the construction process, which includes input representation, feature selection and integration, tree generation, as well as *interpretability and visualisation of our tree-based model*.

**Preliminary Setup** Before delving into the components of the proposed explanation model PEACH, it is crucial to distinguish between the pretrained model, fine-tuned model, and its contextual embedding. Figure 4 in the Appendix illustrates the pretraining and fine-tuning step. The pretraining step involves utilising a large amount of unlabelled data to learn the overall language representation, while the fine-tuning step further refines this knowledge and generates better-contextualised embeddings on task-specific labelled datasets. Fine-tuned contextual pretrained embeddings typically serve as a valuable resource for representing the text classification capabilities of various deep learning models. Therefore, PEACH leverages these contextualised embeddings to explain the potential outcomes of a series of related choices using a contextual and hierarchical decision tree.

## 2.1 Input Embedding Construction

We first wrangle the extracted pretrained contextual embeddings in order to demonstrate the contextual understanding capability of the fine-tuned model. Note that we use fine-tuned contextual embedding as input features by applying the following two steps.

### 2.1.1 Step 1: Fine-Tuning

Given a corpus with $n$ text documents, denoted as $T = \{t_1, t_2, \ldots, t_n\}$, where each $t_a$ represents a document instance from a textual dataset. Note each document can be from any text-based corpus, such as news articles, movie reviews, or medical abstracts, depending on the specific dataset or task. Each document can be represented as a semantic embedding by pretrained models. In order to retrieve the contextual representation, we firstly tokenise $t_a$ for $a \in [1, n]$ with the pretrained model tokeniser and finetune the PLMs on the tokenised contents of all documents, with the goal of predicting the corresponding document label. Then, we extract the $d$-dimensional embedding of the PLMs [CLS] token as the contextualised document embedding $e_a \in E$ for $a \in [1, n]$ ($d = 768$).

### 2.1.2 Step 2: Feature Processing

By using all the embeddings in $E$ as row vectors, we construct a feature matrix $M \in \mathbb{R}^{n*d}$. This feature matrix can be represented as $[c_1 \ c_2 \ \ldots \ c_d]$ where each $c_i$ corresponds to the column feature vector along the $i$-th dimension, which contains embedding values for the $i$-th dimension from all document embeddings. To optimise the utilisation of the embedding feature matrix in decision tree training, we experiment with various feature selection methods, including the following statistical approaches and a deep learning approach, to extract the most informative features.

**Statistical Approaches** We first employ statistical approaches to extract informative features from the column features of $M$. We calculate the **correlation** between each pair of dimensions using the Pearson correlation coefficient. For dimensions $i$ and $j$ ($i, j \in [1, d]$), the correlation $R_{i,j}$ is computed as:

$$R_{i,j} = Pearson(c_i, c_j) \tag{1}$$

for each $i, j \in [1, d]$.

Dimensions with high Pearson correlation values indicate similar semantic features during fine-tuning. To identify those similar dimensions, we use a percentile $v$ to find the correlation threshold. The threshold $t$ is calculated as

$$t = P_v(R) \tag{2}$$

which gives us the $v$-th percentile value in $R$. Using this threshold, we cluster the 768 dimensions and take the average to reduce the number of features we will have eventually.

We divide the set of column features $\{c_1, c_2, \ldots, c_d\}$ into $m$ exclusive clusters. Starting from $c_1$, we find all the dimensions having correlations greater than $t$ with $c_1$ and collect them as a new cluster $C_1$. Among the remaining dimensions, we take the first column feature (e.g. $c_k \notin C_1$) as the new cluster centre and find all the dimensions correlating greater than $t$ with $c_k$, and consider them as the new cluster $C_2$. This process is repeated iteratively until all dimensions are assigned to a certain cluster. In this way, we re-arranged all the dimensions into a set of clusters $\mathcal{C} = \{C_1, C_2, \ldots, C_m\}$ where the column vectors in each cluster have correlations greater than $t$ with the cluster centre.

In addition to Pearson, we explore **K-means Clustering** as an alternative method to cluster the dimension vectors. The K-means aims to minimise the objective function given by:

$$L(M) = \sum_{i=1}^{m} \sum_{j=1}^{d} (||v_i - c_j||)^2 \tag{3}$$

where $v_i$ is the cluster centre for each cluster $C_i$. After clustering, we merge the features in each cluster as a single feature vector by taking their average since they exhibit high correlation. By combining the representations from each cluster, we obtain the final feature matrix $F \in \mathbb{R}^{n*m}$. This successfully reduces the number of features from the original embedding dimension d to the number of clusters $m$. The resulting feature matrix $F$ can be directly used as input for training decision trees using ID3/C4.5/CART algorithms.

**Deep Learning Approach** We also apply a Convolutional Neural Network (CNN) to extract the input feature matrix $F$ from the initial embedding feature matrix $E$. Our CNN consists of two blocks, each comprising a 1D convolutional layer followed by a 1D pooling layer. The network is trained to predict the document class based on the output of the last pooling layer, minimising the cross-entropy loss. The filter of each layer reduces the dimension according to the following way, ensuring the last pooling layer has dimension $m$:

$$D_{out} = \frac{D_{in} - f + 2p}{s} - 1 \tag{4}$$

where $D_{in}$ is the input feature dimension of the convolution/pooling layer, $D_{out}$ is the output feature dimension of the convolution/pooling layer, $f$ is the filter size, $p$ is the padding size, and $s$ is the stride size for moving the filter.

## 2.2 DECISION TREE GENERATION

We apply the feature matrix $F$ to construct various types of decision trees. In this section, we describe several traditional decision tree training algorithms that we adopted for our model. The **ID3** algorithm calculates the information gain to determine the specific feature for splitting the data into subsets. For each input feature $f_i \in F$ that has not been used as a splitting node previously, the information gain (IG) is computed as follows to split the current set of data instances $S$:

$$IG(S, f_i) = H(S) - \sum_{t \in T} p(t)H(t)$$
$$= H(S) - H(S|f_i) \tag{5}$$

where $H(S)$ is entropy of the current set of data $S$, $T$ is the subsets of data instances created from splitting $S$ by $f_i$ and p(t) is the proportion of the number of elements in $t$ to the number of elements in $S$ and $H(t)$ is the entropy of subset $t$. The feature with the maximum information gain is selected to split $S$ into two different splits as the child nodes. The **C4.5** algorithm calculates the gain ratio to select the specific feature to split the data into subsets. It is similar to ID3 but instead of calculating information gain, it calculates the gain ratio to select the splitting feature. The gain ratio is calculated as follows:

$$GainRatio(S, f_i) = \frac{IG(S, f_i)}{SplitInfo(S, f_i)} \tag{6}$$

where

$$SplitInfo(S, f_i) = \sum_{t \in T} p(t)log_2(t) \tag{7}$$

The **CART** algorithm calculates the Gini index to select the specific feature for splitting the data into subsets. The Gini index is defined as:

$$Gini(S, f_i) = 1 - \sum_{x=1}^{n} (P_x)^2 \tag{8}$$

where $P_x$ is the probability of a data instance being classified to a particular class. The feature with the smallest Gini is selected as the splitting node. The **Random Forest (RF)** algorithm functions as an ensemble of multiple decision trees, where each tree is generated using a randomly selected subset of the input features. The individual trees in the forest can be constructed using any of the aforementioned algorithms.

## 2.3 INTERPRETABILITY AND VISUALISATION

As mentioned earlier, PEACH aims to foster global and local interpretability for text classification by arranging hierarchical-structured decision rules. Note that PEACH aims to simulate the context understanding, show the text classification decision-making process and transparently present a hierarchical decision tree. For the decision tree, the leaves present the class distributions, the paths from the root to the leaves represent the learned classification rules, and the nodes contain representative parts of the textual corpus. In this section, we explain the way of representing the node in the tree structure and which way of visualisation presents a valuable pattern for human interpretation.

### 2.3.1 INTERPRETABLE PROTOTYPE NODE

The aim of the decision tree nodes in PEACH is to visualise the context understanding and the most common words in the specific decision path, and simulate the text classification decision-making process. Word clouds are great visual representations of word frequency that give greater prominence to words that appear more frequently in a source text. These particular characteristics of word clouds would be directly aligned with the aim of the decision tree nodes. For each node in the tree, we collect all the documents going through this specific node in their decision path. These documents are converted into lowercase, tokenised, and the stopwords in these documents are removed. Then, Term Frequency Inverse Document Frequency (TFIDF) of the remaining words is calculated and sorted. We take the 100 distinct words with the top TFIDF values to be visualized as a word cloud. This gives us an idea of the semantics that each node of the tree represents and how each decision path evolves before reaching the leaf node (the final class decision).

### 2.3.2 VISUALISATION FILTER

The more valuable visualisation pattern the model presents, the better the human-interpretable models are. We apply two valuable token/word types in order to enhance the quality of visualisation of the word cloud node in the decision tree. First, we adopt Part-of-Speech (PoS) tagging, which takes into account which grammatical group (Noun, Adjective, Adverb, etc.) a word belongs to. With this PoS tagging, it is easy to focus on the important aspect that each benchmark has. For example, the sentiment analysis dataset would consider more emotions or polarities so adjective or adverb-based visualisation would be more valuable. Secondly, we also apply a Named Entity Recognition (NER), which is one of the most common information extraction techniques and identifies relevant nouns (person, places, organisations, etc.) from documents or corpus. NER would be a great filter for extracting valuable entities and the main topic of the decision tree decision-making process.

## 3 EVALUATION SETUP[2]

### 3.1 DATASETS[3]

We evaluate PEACH with 5 state-of-the-art PLMs on 9 benchmark datasets. Those datasets encompass five text classification tasks, including Natural Language Inference (NLI), Sentiment Analysis (SA), News Classification (NC), Topic Analysis (TA), and Question Type Classification (QC). **1) Natural Language Inference (NLI)** *Microsoft Research Paraphrase (MSRP)* (Dolan et al., 2004) contains 5801 sentence pairs with binary labels. The task is to determine whether each pair is a paraphrase or not. The training set contains 4076 sentence pairs and 1725 testing pairs for generating decision trees. During PLM finetuning, we randomly split the training set with a 9:1 ratio so 3668 pairs are used for training and 408 pairs are used for validation. *Sentences Involving Compositional Knowledge (SICK)* (Marelli et al., 2014) dataset consists of 9840 sentence pairs that involve compositional semantics. Each pair can be classified into three classes: entailment, neutral or contradiction. The dataset has 4439, 495, and 4906 pairs for training, validation and testing sets. For both MSRP and SICK, We combine each sentence pair as one instance when finetuning PLMs. **2) Sentiment Analysis (SA)** The sentiment analysis datasets in this study are binary for predicting positive or negative movie reviews. *Standford Sentiment Treebank (SST2)* (Socher et al., 2013) has 6920, 872 and 1821 documents for training, validation and testing. The *MR* (Pang & Lee, 2005)

---

[2]The baseline description can be found in the Appendix B

[3]The statistics can be found in the Table 4

Table 1: Classification performance comparison between fine-tuned contextual embeddings and those with PEACH. Best performances among the baselines are underscored, and the best performances among our PEACH variants are bolded.

| Model | MSRP | SST2 | MR | IMDB | SICK | BBCNews | TREC | 20ng | Ohsumed |
|---|---|---|---|---|---|---|---|---|---|
| BERT (Devlin et al., 2019) | 0.819 | 0.909 | 0.857 | 0.870 | 0.853 | 0.969 | 0.870 | 0.855 | 0.658 |
| RoBERTa (Liu et al., 2020) | 0.824 | 0.932 | 0.867 | 0.892 | 0.881 | 0.959 | 0.972 | 0.802 | 0.655 |
| ALBERT (Lan et al., 2020) | 0.665 | 0.907 | 0.821 | 0.879 | 0.838 | 0.917 | 0.938 | 0.794 | 0.518 |
| XLNet (Yang et al., 2019) | 0.819 | 0.907 | 0.907 | 0.905 | 0.727 | 0.959 | 0.950 | 0.799 | 0.669 |
| ELMo (Peters et al., 2018) | 0.690 | 0.806 | 0.751 | 0.804 | 0.608 | 0.845 | 0.790 | 0.537 | 0.359 |
| PEACH (BERT) | 0.816 | 0.885 | 0.853 | 0.871 | 0.837 | **0.975** | **0.978** | **0.849** | 0.649 |
| PEACH (RoBERTa) | **0.819** | **0.938** | **0.872** | **0.893** | **0.877** | 0.947 | 0.968 | 0.800 | **0.651** |
| PEACH (ALBERT) | 0.650 | 0.885 | 0.804 | 0.878 | 0.834 | 0.921 | 0.942 | 0.783 | 0.497 |
| PEACH (XLNet) | 0.809 | 0.903 | 0.790 | **0.899** | 0.832 | 0.964 | 0.966 | 0.776 | 0.638 |
| PEACH (ELMo) | 0.638 | 0.632 | 0.510 | 0.725 | 0.565 | 0.900 | 0.702 | 0.164 | 0.284 |

Table 2: The effects of feature processing approach.

| Model | MSRP | SST2 | MR | IMDB | SICK | BBCNews | TREC | 20ng | Ohsumed |
|---|---|---|---|---|---|---|---|---|---|
| PEACH (Pearson) | 0.817 | 0.913 | 0.862 | 0.892 | 0.867 | 0.972 | **0.978** | 0.845 | 0.645 |
| PEACH (K-means) | **0.819** | **0.938** | 0.869 | **0.893** | **0.877** | **0.975** | 0.974 | **0.849** | **0.651** |
| PEACH (CNN) | 0.817 | 0.936 | **0.872** | 0.890 | 0.870 | 0.974 | 0.974 | 0.801 | 0.621 |

has 7108 training and 3554 training documents. *IMDB* (Maas et al., 2011) and 25000 training and 25000 training documents. For MR and IMDB, since no official validation split is provided, the training sets are randomly split into a 9:1 ratio to obtain a validation set for finetuning PLMs. **3) News Classification (NC)** The *BBCNews* is used to classify news articles into five categories: entertainment, technology, politics, business, and sports. There are 1225 training and 1000 testing instances, and we further split the 1225 training instances with 9:1 ratio to get the validation set for finetuning PLMs. *20ng* is for news categorization with 11314 training and 7532 testing documents and aims to classify news articles into 20 different categories. Similar to BBCNews, the training set is split with 9:1 ratio to obtain the finetuning validation set. **4) Topic Analysis (TA)** The *Ohsumed* provides 7400 medical abstracts, with 3357 train and 4043 test, to categorise into 23 disease types. **5) Question Type Classification (QC)** The *Text REtrieval Conference (TREC)* Question Classification dataset offers 5452 questions in the training and 500 questions in the testing set. This dataset categorises different natural language questions into six types: abbreviation, entity, description and abstract, human, location, and numbers.

## 3.2 IMPLEMENTATION DETAILS

We perform finetuning of the PLMs and then construct the decision tree-based text classification model for the evaluation. We initialise the weights from **five base models: bert-base-uncased, roberta-base, albert-base-v2, xlnet-base-cased and the original 93.6M ELMo** for BERT, RoBERTa, ALBERT, XLNet, and ELMo respectively. A batch size of 32 is applied for all models and datasets. The learning rate is set to 5e-5 for all models, except for the ALBERT model of SST2, 20ng and IMDB datasets, where is set to 1e-5. All models are fine-tuned for 4 epochs, except for the 20ng, which used 30 epochs. To extract the features from learned embedding and reduce the number of input features into the decision tree, we experiment with quantile thresholds of 0.9 and 0.95 for correlation methods. For k-means clustering, we search for the number of clusters from 10 to 100 (step size: 10 or 20), except for IMDB where we search from 130 to 220 (step size: 30). For CNN features, we use kernel size 2, stride size 2, and padding size 0 for two convolution layers. The same hyperparameters are applied to the first pooling layer, except for IMDB where stride size 1 is used to ensure we can have enough input features for the next convolutional block and get enough large number of features from the last pooling layer. Kernel size and stride size for the last pooling layer are adjusted to maintain consistency with the number of clusters used in k-means clustering. Decision trees are trained using the chefboost library with a maximum depth of 95. We used em_core_web_sm model provided by spaCy library to obtain the NER and POS tags for visualization filters. All experiments are conducted on Google Colab with Intel(R) Xeon(R) CPU and NVIDIA T4 Tensor Core GPU. Classification accuracy is reported for comparison.

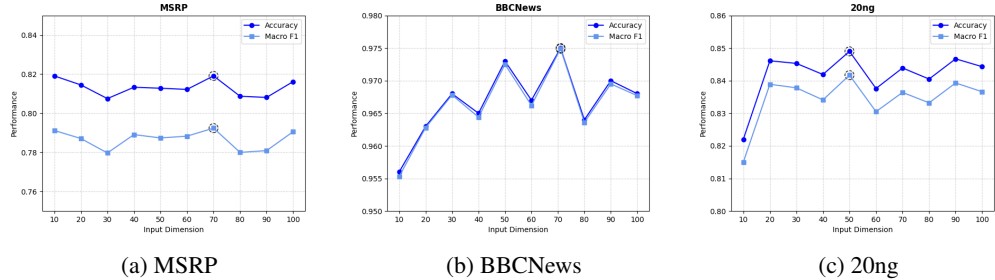

Figure 2: The effects of input feature dimension with PEACH on three datasets, including MSRP, BBCNews, and 20ng

## 4 RESULTS

### 4.1 OVERALL CLASSIFICATION PERFORMANCE

We first evaluate the utility of PLMs after incorporating PEACH in Table 1. As shown in the table, our proposed PEACH with PLMs (rows 7-11) outperforms or performs similarly to the fine-tuned PLMs (rows 2-6) in all benchmark datasets. It is worth noting that our model outperforms the baselines in all binary sentiment analysis datasets (SST2, MR, IMDB) when utilising RoBERTa features in our PEACH model. Furthermore, our model outperforms the baselines in more general domain datasets with multiple classes, such as BBC News, TREC, and 20ng, when BERT features are employed in PEACH. We also experimented with various types of PLMs, other than BERT and RoBERTa, on our PEACH, including BERT-based (ALBERT), generative-based (XLNet), and LSTM-based model (ELMo). In general, RoBERTa and BERT performed better across all datasets, except IMDB, where XLNet and its PEACH model showed superior performance. This observation is attributed to the larger corpus of IMDB, which requires more features for an accurate explanation.

### 4.2 ABLATION AND PARAMETER STUDIES[4]

**The effects of Feature Processing** All three feature processing methods we propose in Section 2.1.2 work well overall. Table 2 shows that K-means demonstrated the best across most datasets, except for MR and TREC. CNN worked better on the MR dataset. Conversely, Pearson correlation grouping worked better for TREC. The fine-tuned BERT model captured features that exhibited stronger correlations with each other rather than clustering similar question types together.

**The effects of Input Dimension** We then evaluate the effects of input feature dimension size with PEACH. We selected three datasets, including BBCNews (the largest average document length with a small data size), 20ng (a large number of classes with a larger dataset size) and MSRP (a moderate number of documents and a moderate average length). We present macro F1 and Accuracy to analyse the effects of class imbalance for some datasets. Figure 2 shows there is no difference in the trend for F1 and the trend for accuracy on these datasets; especially the relatively balanced dataset BBCNews provides almost identical F1 and accuracy. The binary dataset MSRP does not lead to large gaps in the performance of different input dimensions, however, for those with more classes (BBCNews and 20ng), there is a noticeable performance drop when using extremely small input dimensions like 10. The performance difference becomes less significant as the input dimension increases beyond 20.

**The Maximum Tree Depth Analysis** was conducted to evaluate the visualisation of the decision-making pattern. The result can be found in Appendix C.

---

[4]The effects of the Decision Tree is in Appendix A

Table 3: Human evaluation results. Pairwise comparison between PEACH with LIME and Anchor across the interpretability. The 'Agree' column shows the Fleiss' Kappa results.

| MR | | | | SST2 | | | | TREC | | | | BBCNews | | | |
|---|---|---|---|---|---|---|---|---|---|---|---|---|---|---|---|
| PEACH | LIME | Tie | Agree | PEACH | LIME | Tie | Agree | PEACH | LIME | Tie | Agree | PEACH | LIME | Tie | Agree |
| 92.3 | 1.3 | 6.4 | 0.83 | 88.4 | 1.8 | 9.8 | 0.78 | 96.1 | 1.2 | 2.7 | 0.81 | 84.6 | 3.9 | 11.5 | 0.75 |
| PEACH | Anchor | Tie | Agree | PEACH | Anchor | Tie | Agree | PEACH | Anchor | Tie | Agree | PEACH | Anchor | Tie | Agree |
| 94.6 | 2.1 | 3.3 | 0.85 | 87.2 | 2.5 | 10.3 | 0.76 | 95.4 | 1.8 | 2.8 | 0.83 | 85.3 | 3.5 | 11.2 | 0.71 |

### 4.3 HUMAN EVALUATION: INTERPRETABILITY AND TRUSTABILITY

To assess interpretability and trustability, we conducted a human evaluation[5], 26 human judges[6] annotated 75 samples from MR, SST2, TREC, and BBC News. Judges conducted the pairwise comparison between local and global explanations generated by PEACH against two commonly used text classification-based interpretability models, LIME (Ribeiro et al., 2016) and Anchor (Ribeiro et al., 2018). The judges were asked to choose the approach they trusted more based on the interpretation and visualisation provided. We specifically considered samples, where both LIME and PEACH or Anchor and PEACH predictions were the same, following Wan et al. (2021). Among 26 judges, our PEACH explanation evidently outperforms the baselines by a large margin. All percentages in the first column of all four datasets are over 84%, indicating that the majority of annotators selected our model to be better across interpretability and trustability. The last column ('Agree') represents results from the Fleiss' kappa test used to assess inter-rater consistency (Fleiss, 1971), and all the agreement scores are over 0.7 which shows a strong level of agreement between annotators. Several judges commented that the visualisation method of PEACH allows them to see the full view of the decision-making process in a hierarchical decision path and check how the context is trained in each decision node. This indicates a higher level of trust in PEACH than in the saliency technique commonly employed in NLP.

## 5 ANALYSIS AND APPLICATION

**Visualisation Analysis** Figure 1 in Section 1 shows the sample interpretations by PEACH on MR, a binary dataset predicting positive or negative movie reviews. The global explanation in Figure 1(left) faithfully shows the entire classification decision-making behaviour in detail. Globally, the decision tree and its nodes cover various movie-related entities and emotions, like DOCUMENTARY, FUN, FUNNY, ROMANTIC, CAST, HOLLYWOOD, etc. In addition, final nodes (leaves) are passed via the clear path with definite semantic words, distinguishing between negative and positive reviews. In addition to the global interpretation, our PEACH can produce a local explanation (Figure 1, right). By applying generated decision rules to the specific input text, a rule path for the given input simulates the decision path that can be derived to the final classification (positive or negative).

We also present how PEACH can visualise the interpretation by comparing the successful and unsuccessful pretrained embeddings in Figure 3. While the successful one (RoBERTa with MR - Figure 3 left) shows a clear and traceable view of how they can classify the positive samples by using adjectives like moving and engaging, the unsuccessful one (ELMo with MR - Figure 3 right) has many ambiguous terminologies and does not seem to understand the pattern in both positive and negative classes, e.g. amusing is in the negative class. More visualisations on different datasets and models are shown in Appendix D and E.

**Application: PEACH** We developed an interactive decision tree-based text classification decision-making interpretation system for different PLMs. The user interface and detailed description are in Appendix F and Figure 5.

---

[5]Sample Cases for the Human Evaluation is in the Appendix Figure 26

[6]Annotators are undergraduates and graduates in computer science; 6 females and 20 males. The number of human judges and samples is relatively higher than other NLP interpretation papers (Rajagopal et al., 2021)

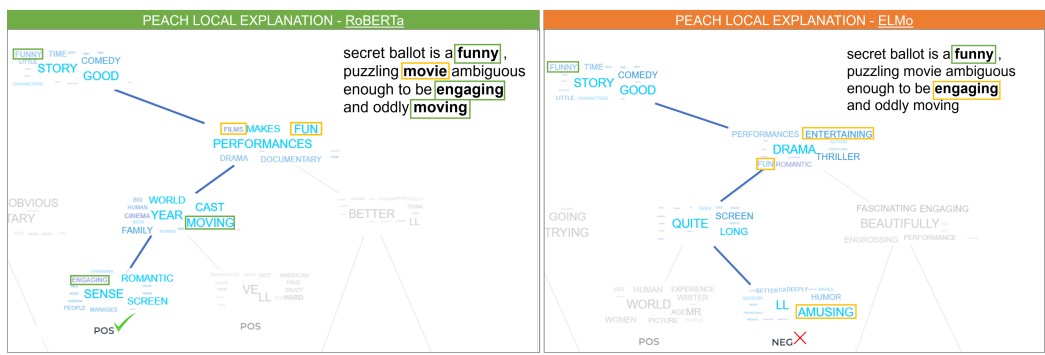

Figure 3: The local explanation decision trees generated for a **positive** review in MR dataset based on fine-tuned RoBERTa embedding compared to fine-tuned ELMo embedding.

## 6 RELATED WORKS

**Interpretable Models in NLP** Among interpretable NLP models, feature attribution-based methods are the most common[7]. There are mainly four types, including Rationale Extraction (Lei et al., 2016; Bastings et al., 2019; Yu et al., 2019; Sha et al., 2021), Input Perturbation (Ribeiro et al., 2016; 2018; Feng et al., 2018; Slack et al., 2020), Attention Methods (Luo et al., 2018; Mao et al., 2019), and Attribution Methods (He et al., 2019; Du et al., 2019). Such models locally explain the prediction based on the relevance of input features (words). However, global explainability is crucial to determine how much each feature contributes to the model's predictions of overall data. A few recent studies touched on the global explanation idea and claimed they have global interpretation by providing the most relevant concept (Rajagopal et al., 2021) or the most influential examples (Han et al., 2020) searched from the corpus to understand why the model made certain predictions. However, such global explanations do not present the overall decision-making flow of the models.

**Tree-structured Model Interpretation** The recent studies adopting decision-tree into neural networks (Irsoy et al., 2012; Zharmagambetov & Pak, 2015; Humbird et al., 2019; Frosst & Hinton, 2018; Fuhl et al., 2020; Lee & Jaakkola, 2020; Wang et al., 2020; Tanno et al., 2019) introduced neural trees compatible with the state-of-the-art of CV and NLP downstream tasks. Such models have limited interpretation or are only suitable to the small-sized dataset. Tree-structured neural models have also been adopted in syntactic or semantic parsing (Shen et al., 2019; Cheng et al., 2018; Le et al., 2018; Nguyen et al., 2020; Wang* et al., 2020; Zhou et al., 2018b; Dong et al., 2019; Yu et al., 2021; Zhang et al., 2021). Few decision-tree-based approaches show the global and local explanations of black-boxed neural models. NBDT (Wan et al., 2021) applies a sequentially interpretable neural tree and uses parameters induced from trained CNNs and requires WordNet to establish the interpretable tree. ProtoTree (Nauta et al., 2021) and ViT-NeT (Kim et al., 2022) construct interpretable decision trees for visualising decision-making with prototypes for CV applications. Despite their promise, those have not yet been adopted in NLP field.

## 7 CONCLUSION

This study introduces PEACH, a novel tree-based explanation technique for text-based classification using pretrained contextual embeddings. While many NLP applications rely on PLMs, the focus has often been on employing them without thoroughly analysing their contextual understanding for specific tasks. PEACH addresses this gap by providing a human-interpretable explanation of how text-based documents are classified, using any pretrained contextual embeddings in a hierarchical tree-based manner. The human evaluation also indicates that the visualisation method of PEACH allows them to see the full global view of the decision-making process in a hierarchical decision path, making fine-tuned PLMs interpretable. We hope that the proposed PEACH can open avenues for understanding the reasons behind the effectiveness of PLMs in NLP.

---

[7]Some studies cover the language explanation-based, probing-based or counterfactual explanation-based methods, but the text-based model interpretation methods are dominant by feature attribution-based approaches.

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

## A  THE EFFECTS OF DECISION TREE ALGORITHM

In order to evaluate the effect of the decision tree algorithm to generate the tree for PEACH, we tested several decision tree algorithms, including ID3, C4.5, CART, and those with Random Forest. Table 5 shows that the overall performance between different decision tree algorithms is very similar to each other. Generally, the Random forests worked better than single trees most of the time, except for SST2 and BBCNews. However, the Random Forest is too crowded and too many rules in the PEACH decision tree output. Hence, it is better to visualise the decision-making path of the specific PLMs in a single decision tree.

Table 4: Detailed Dataset Statistics

| Method | MSRP | SST2 | MR | IMDB | SICK | BBCNews | TREC | 20ng | Ohsumed |
|---|---|---|---|---|---|---|---|---|---|
| # Class | 2 | 2 | 2 | 2 | 3 | 5 | 6 | 20 | 23 |
| # Docs | 5801 | 8741 | 10662 | 50000 | 9345 | 2225 | 5952 | 18846 | 7400 |
| # Train Docs | 4076(70.3%) | 6920(79.2%) | 7108(66.7%) | 25000(50%) | 4439(47.5%) | 1225(55.1%) | 5452(91.6%) | 11314(60.0%) | 3357(45.4%) |
| # Test Docs | 1725(29.7%) | 1821(20.8%) | 3554(33.3%) | 25000(50%) | 4906(52.5%) | 1000(44.9%) | 500(8.4%) | 7532(40.0%) | 4043(54.6%) |
| # Words | 17873 | 15481 | 18334 | 181061 | 2318 | 32772 | 8900 | 42106 | 14127 |
| Avg Length | 37.7 | 17.5 | 19.4 | 230.3 | 19.3 | 388.3 | 8.7 | 189.0 | 121.6 |
| Task | NLI | SA | SA | SA | NLI | NC | QC | NC | TA |

Table 5: The effects of Decision Tree Algorithm

| Method | MSRP | SST2 | MR | IMDB | SICK | BBCNews | TREC | 20ng | Ohsumed |
|---|---|---|---|---|---|---|---|---|---|
| ID3 | 0.797 | 0.931 | 0.860 | 0.874 | 0.848 | 0.969 | 0.958 | 0.815 | 0.561 |
| C4.5 | 0.796 | 0.925 | 0.856 | 0.884 | 0.847 | 0.963 | 0.956 | 0.810 | 0.566 |
| CART | 0.784 | **0.938** | 0.856 | 0.871 | 0.851 | **0.975** | 0.960 | 0.813 | 0.561 |
| RF(ID3) | 0.809 | 0.935 | 0.868 | 0.891 | 0.872 | 0.970 | 0.968 | 0.845 | **0.651** |
| RF(C4.5) | 0.814 | 0.934 | **0.872** | 0.891 | **0.877** | 0.968 | 0.966 | **0.849** | 0.649 |
| RF(CART) | **0.819** | 0.936 | 0.864 | **0.893** | 0.870 | 0.963 | **0.978** | 0.841 | 0.640 |
| Rule number (best single tree) | 341 | 38 | 176 | 702 | 244 | 7 | 92 | 119 | 616 |
| Max depth (best one) | 22 | 11 | 17 | 30 | 25 | 5 | 9 | 12 | 11 |
| Max depth (best single tree) | 29 | 11 | 18 | 95 | 18 | 5 | 12 | 10 | 80 |
| Max depth (best forest) | 22 | 6 | 17 | 30 | 25 | 3 | 9 | 12 | 11 |
| Input dimension) | 70 | 70 | 70 | 220 | 90 | 71 | 31 | 50 | 80 |
| Tree number | 5 | 1 | 5 | 5 | 5 | 1 | 5 | 10 | 10 |

## B  BASELINE

For PEACH with the fine-tuned PLMs, we compare ours with the original PLMs. As our PEACH involves finetuning pretrained models to extract embedding features, we finetune a linear classification layer based on the [CLS] token of the pretrained language models, and report the results on the fine-tuned BERT (Devlin et al., 2019), RoBERTa (Liu et al., 2020), ALBERT (Lan et al., 2020), XLNet(Yang et al., 2019) and ELMo(Peters et al., 2018) models to compare with our PEACH.

## C  MAXIMUM DEPTH ANALYSIS

To visualise the decision-making pattern clearly, we limit the maximum depth of the generated tree. However, it is crucial to keep the classification performance and behaviour similar to the original fine-tuned contextual embedding. Table 6 shows that increasing the tree depth from 3 to 15 has little effect on binary datasets. However, for datasets with multiple classes, decreasing the maximum depth leads to a significant decrease in performance. Insufficient rules to cover all classes, especially at a depth of 3, greatly affect these datasets. Adequate rule coverage is crucial for effectively handling datasets with more classes.

## D  GLOBAL INTERPRETATION AND LOCAL EXPLANATION VISUALISATION AND ANALYSIS

In Section 5, we conducted the qualitative analysis of the interpretation and visualisation. In this Appendix, we would like to present more cases on different datasets, including BBC News, MR,

Table 6: The effects of maximum depth on PEACH. The following tests are conducted on the best setting for each dataset. For MSRP, RoBERTa embedding and K-means clustering is applied to train the decision tree with different maximum depth limit; RoBERTa embedding with CNN is applied for MR; BERT embedding with K-means is applied for BBCNews and 20ng.

| Depth | MSRP | MR | BBCNews | 20ng |
|---|---|---|---|---|
| 3 | 0.825 | 0.869 | 0.806 | 0.492 |
| 5 | 0.824 | 0.866 | 0.975 | 0.706 |
| 10 | 0.823 | 0.869 | 0.975 | 0.841 |
| 15 | 0.834 | 0.867 | 0.975 | 0.849 |

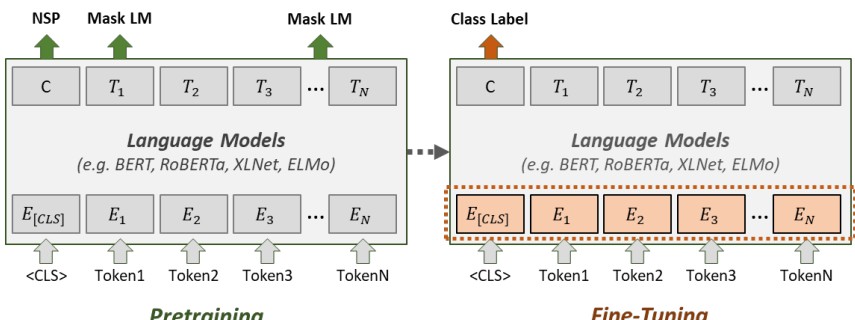

Figure 4: Architecture of pretrained model and fine-tuning process. We extracted the fine-tuned contextual embedding (orange-coloured) as an input of PEACH

and SST. Figures 6 and 7 show the global interpretation decision tree generated on BBC News, while Figures 8, 9 and 10, 11, 12, 13 presents the local explanation for each test document. The figure captions explain the detailed information for each figure. For SST, the global interpretations are in Figures 20 and 21, and the local explanations are in Figures 22 and 23. For MR, the global interpretations are in Figures 14 and 15, and the local explanations are in Figures 16, 17, 18, 19. The important points and remarkable patterns are described in the captions for each figure.

## E    LOCAL EXPLANATION COMPARISON ON PLMs

In Section 5, we conducted the qualitative analysis of the interpretation and visualisation. In this Appendix, we would like to show some examples presented in 24 and 25 to compare how PEACH explained the best-performing PLM and the worst performing PLM for BBCNews and MR. The figure captions explain the detailed information for each figure.

## F    APPLICATION USER INTERFACE

As shown in Figure 5, we developed an interactive decision tree-based text classification decision-making interpretation system for different PLMs. The application would be helpful for anyone who would like to apply PLMs in their text-based prediction/classification tasks. The followings describe the main components of the developed PEACH application. The PEACH application is developed by Python, CSS, and JQuery with the Flask environment. The application has four main components: 1) Dataset Navigator, 2) Parameter Visualisation Panel, 3) Decision Tree Visualisation, and 4) Visualisation Filter.

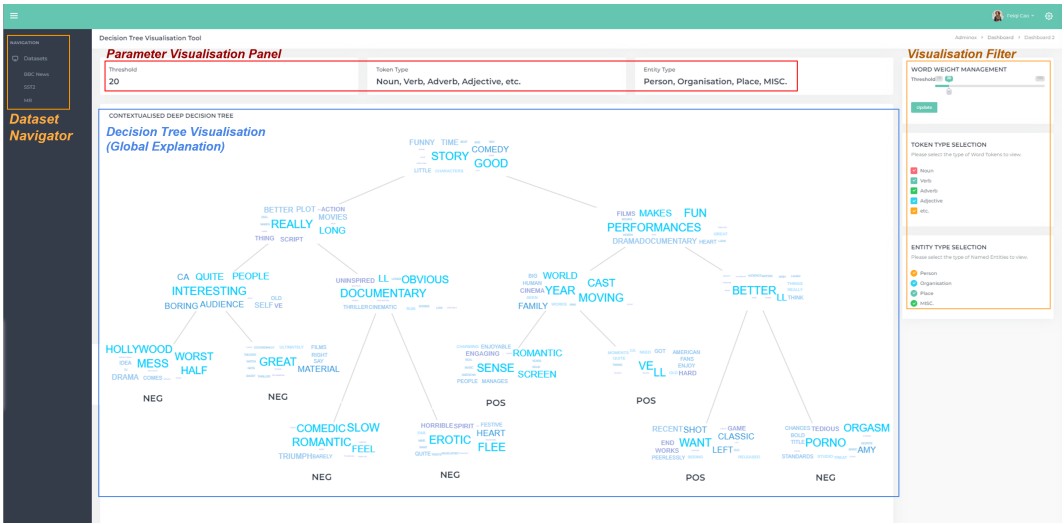

Figure 5: User Interface of the Application PEACH

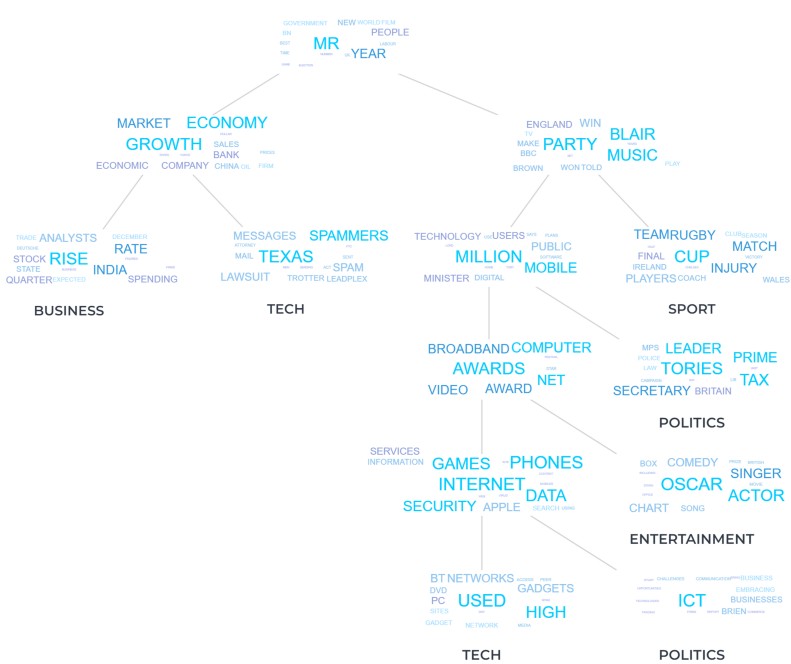

Figure 6: The global explanation decision tree generated on BBCNews dataset based on fine-tuned BERT embedding, where the prototype nodes show the decision path based on the global context for each class. We can observe that Technology news and Business news are grouped together in the first step of the reasoning, before further distinguishing between them. This is probably because there are quite some new articles related to technological corporates, making those two classes share some commonality in the concepts.

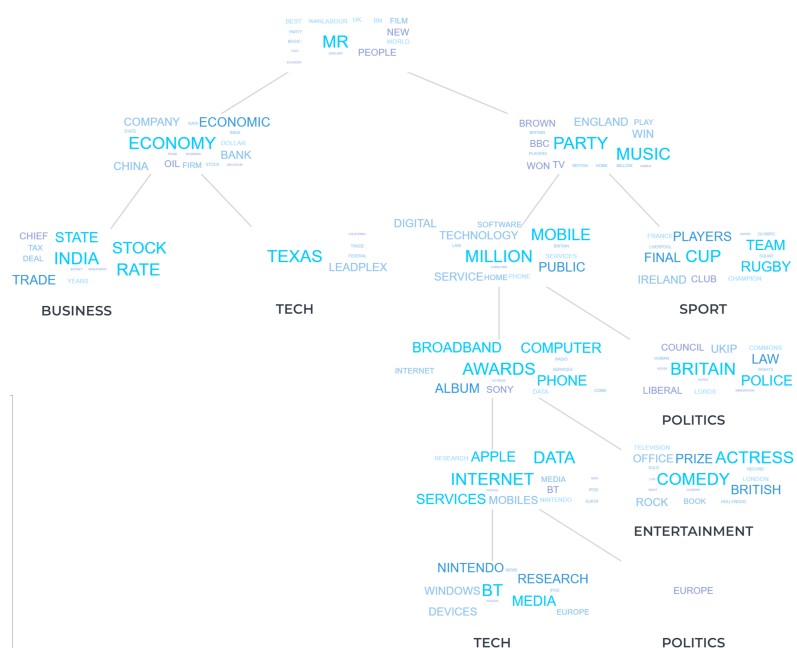

Figure 7: The global explanation decision tree generated on BBCNews dataset based on fine-tuned BERT embedding, with NER filters to display only words with NER tags for organizations and locations, labelled by spaCy library. We can see some country names, sports team names, and company names are coming out more in the prototype nodes to distinguish different types of news. Errors inherited from the NER tagging model will lead to some non-location or non-organisation concepts remaining in the filtered global tree.

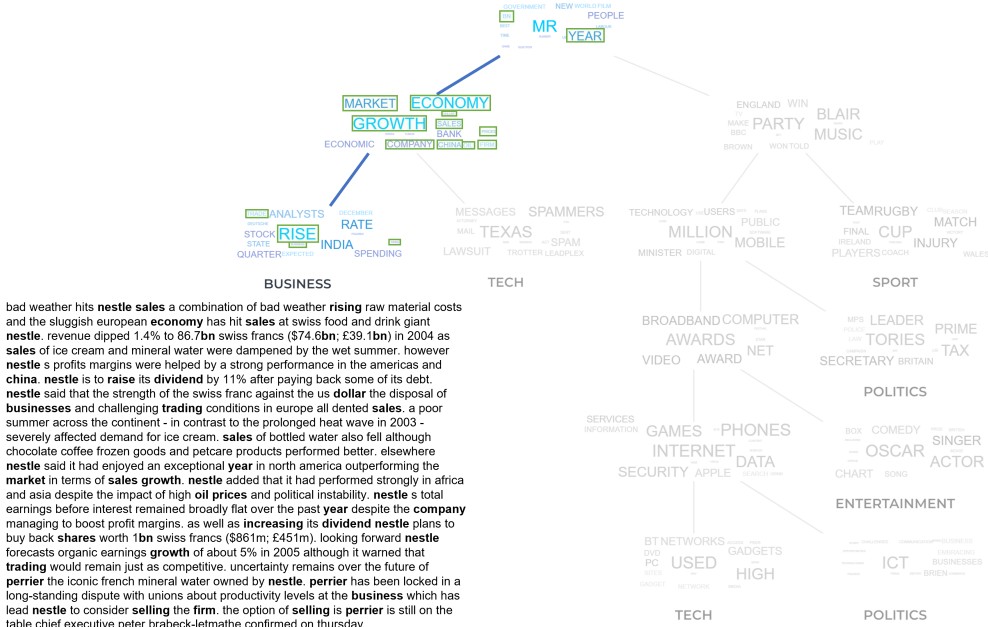

Figure 8: The local explanation decision tree generated for a correctly predicted **Business** news article in BBCNews dataset based on fine-tuned BERT embedding. We can observe that lots of business-related concepts or keywords can be matched from the global trend decision path (highlighted with green squares) to this specific article (highlighted as bolded words).

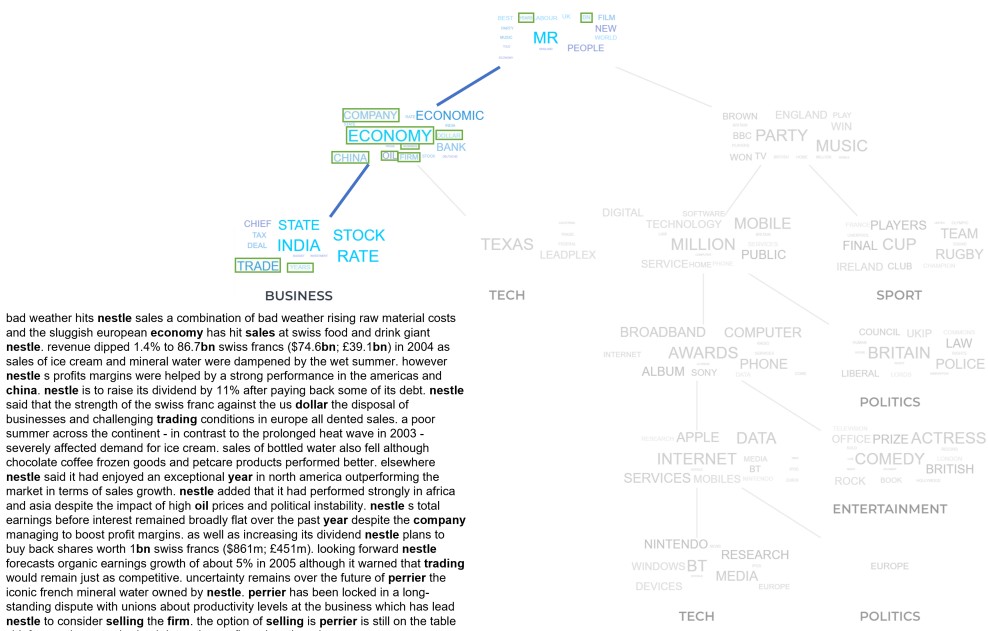

Figure 9: The local explanation decision tree generated for a correctly predicted **Business** news article in BBCNews dataset based on fine-tuned BERT embedding, with NER filters to display only words with NER tags for organizations and locations, labelled by the spaCy library. Business-related concepts or keywords can be matched from the global trend decision path (highlighted with green squares) to this specific article (highlighted as bolded words).

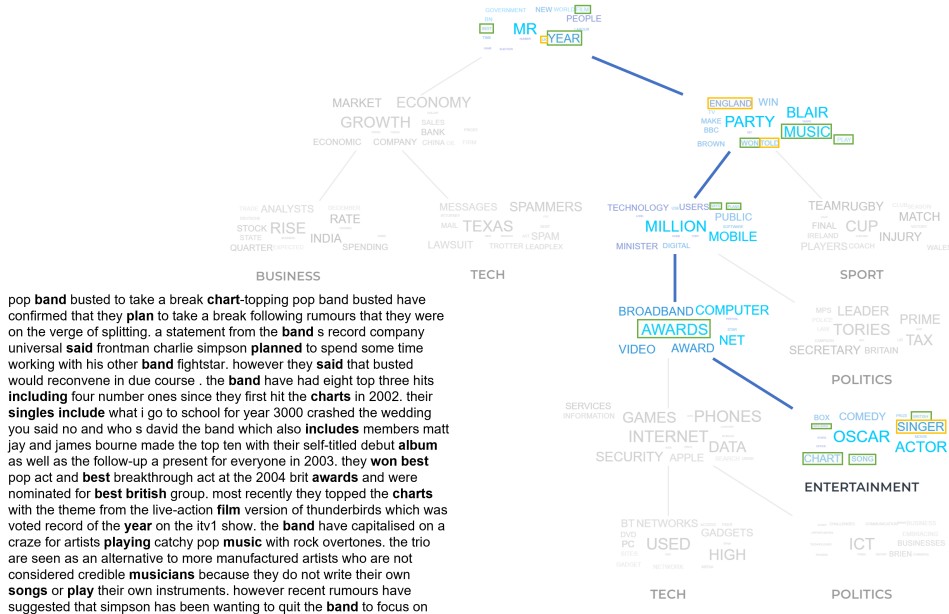

Figure 10: The local explanation decision tree generated for a correctly predicted **Entertainment** news article in BBCNews dataset based on fine-tuned BERT embedding. We can find music-related concepts in this specific news article (highlighted as bolded words), which also come out in the decision path, either as exact matching (highlighted with green squares) or similar concepts like *musician* vs *music* (highlighted with yellow squares).

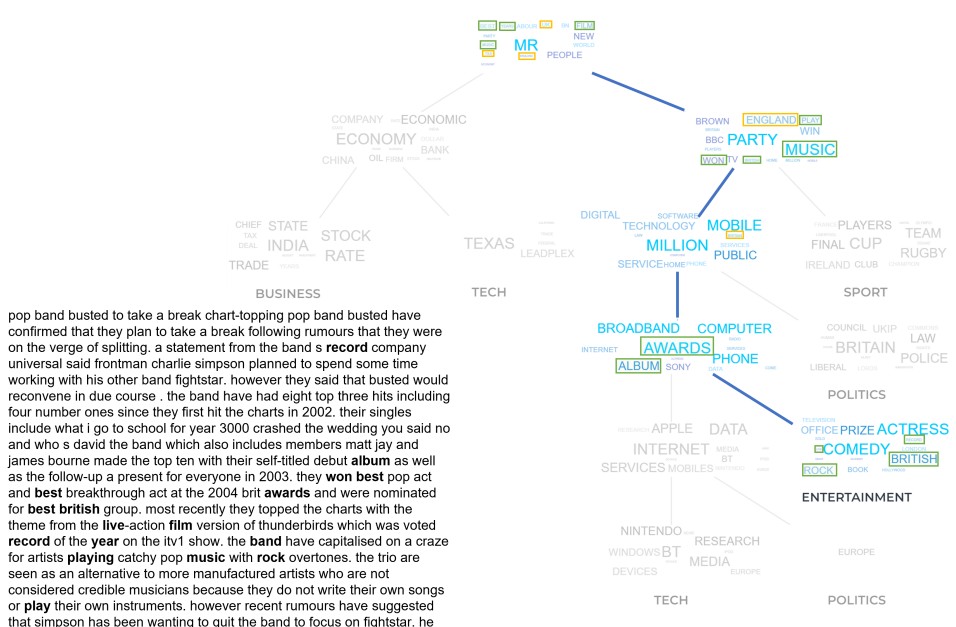

Figure 11: The local explanation decision tree generated for a correctly predicted **Entertainment** news article in BBCNews dataset based on fine-tuned BERT embedding, with NER filters to display only words with NER tags for organizations and locations, labelled by the spaCy library. Not many organization or location names are coming out in this music-related article, as well as the decision path, indicating that the other aspects can be further investigated for more obvious patterns.

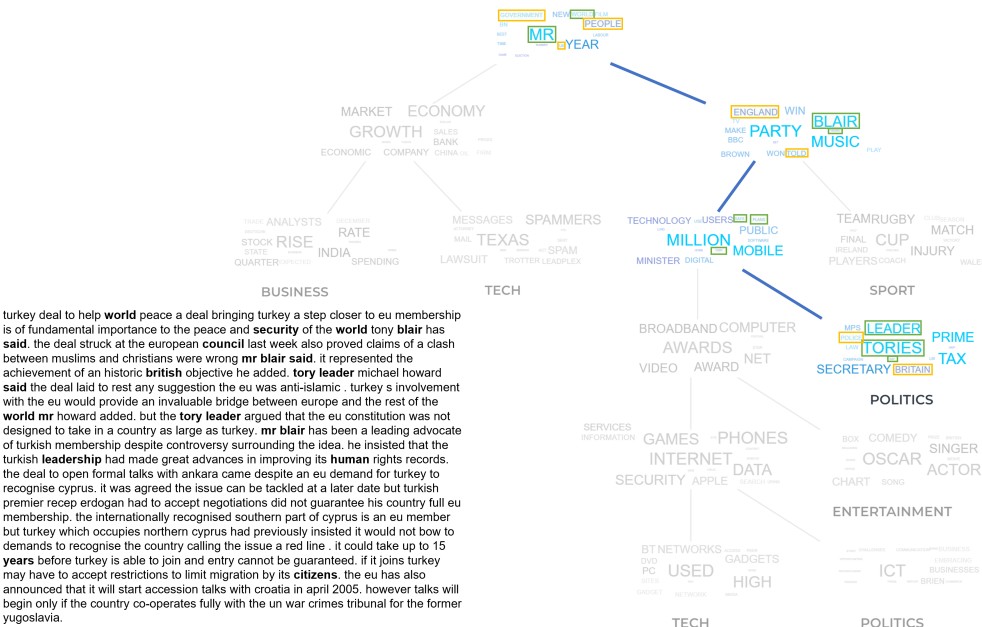

Figure 12: The local explanation decision tree generated for a correctly predicted **Politics** news article in BBCNews dataset based on fine-tuned BERT embedding. We can observe that government people's names or positions from the text (highlighted as bolded text) can be mapped to the decision path (highlighted as green squares for exact mapping, yellow squares for similar concepts).

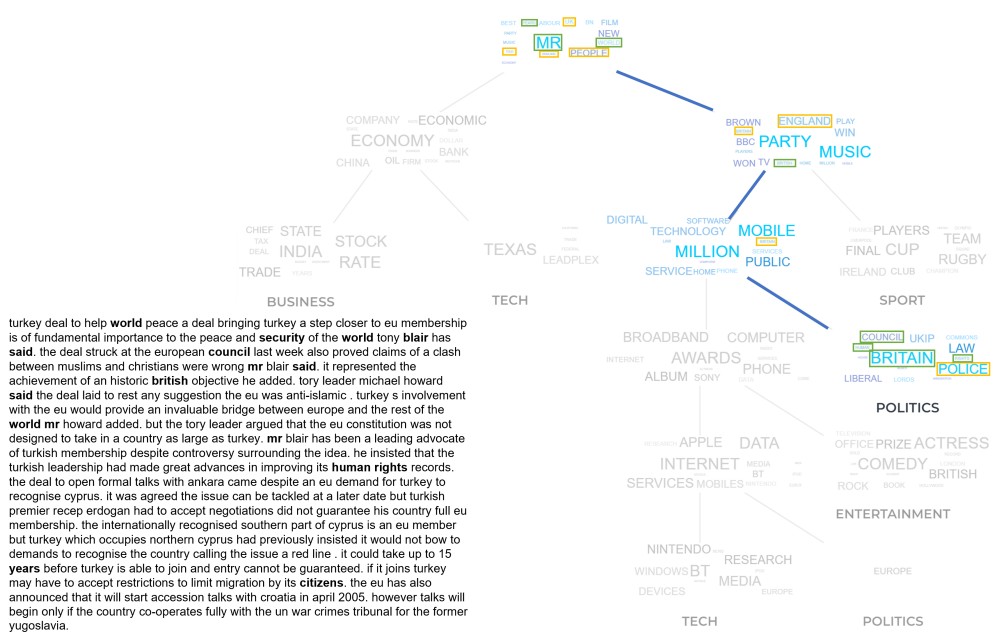

Figure 13: The local explanation decision tree generated for a correctly predicted **Politics** news article in BBCNews dataset based on fine-tuned BERT embedding, with NER filters to display only words with NER tags for organizations and locations, labelled by the spaCy library. In this case, important countries and government departments stand out more in the decision in the presence of the NER filter.

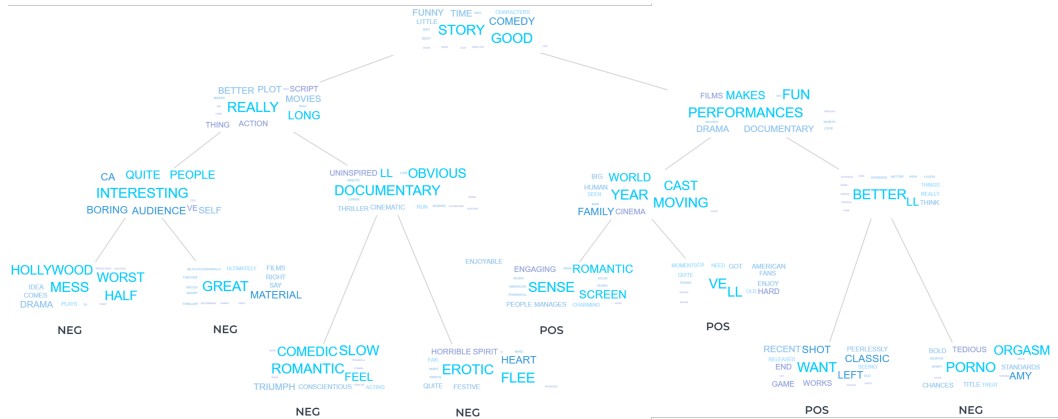

Figure 14: The global explanation decision tree generated on MR dataset based on fine-tuned RoBERTa embedding. Please take note that we explicitly limit the maximum depth of the tree during the training to prevent further branching out beyond 3rd children's level for better performance as well as interpretability, therefore the two children with the same class can be present. We can see words with negative connotations in human understanding tend to appear more in the prototype nodes or decision paths for the negative class, and similarly for the positive class.

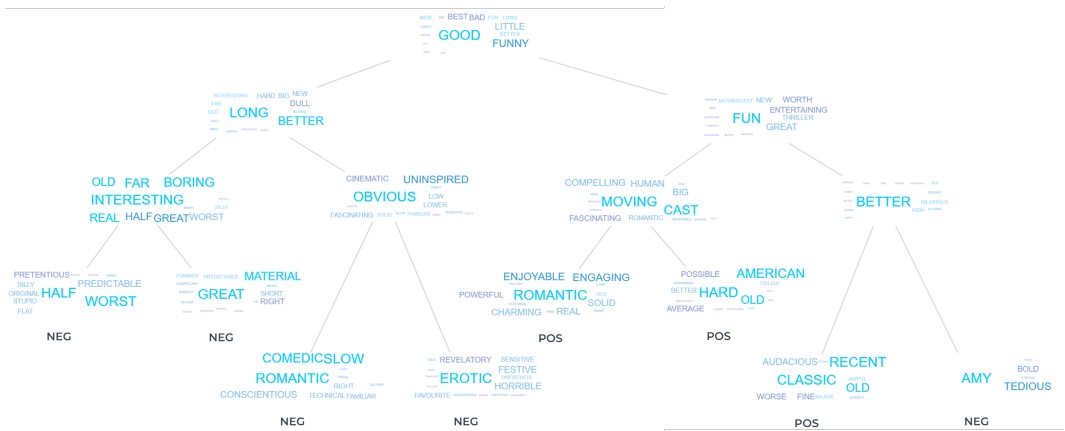

Figure 15: The global explanation decision tree generated on MR dataset based on fine-tuned RoBERTa embedding, with the POS filter to display only words which can be labelled as adjectives by the spaCy library. People tend to use adjectives when giving out movie reviews so investigating the adjectives for this dataset shows a clear pattern for classifying positive or negative reviews. Errors inherited from the POS tagging model will lead to some non-adjectives remaining in the filtered global tree.

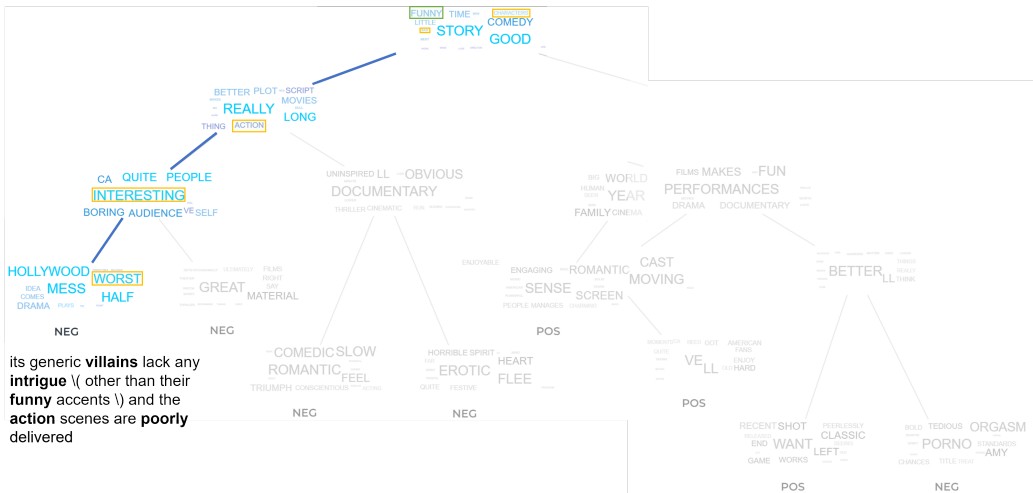

Figure 16: The local explanation decision tree generated for a correctly predicted **Negative** review in MR dataset based on fine-tuned RoBERTa embedding. Concepts related to judging whether something is interesting or not can be found in the decision path and negative word *worst* similar to the concept of *poorly* can be found as well.

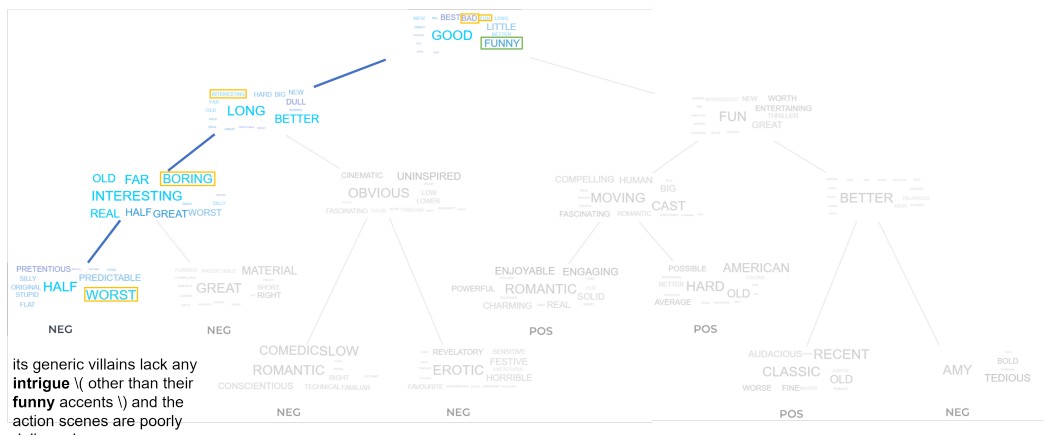

Figure 17: The local explanation decision tree generated for a correctly predicted **Negative** review in MR dataset based on fine-tuned RoBERTa embedding, with the POS filter to display only words which can be labelled as adjectives by the spaCy library. We can see concepts related to the *cast* or *action* is no longer found, leaving only important adjectives for decision-making.

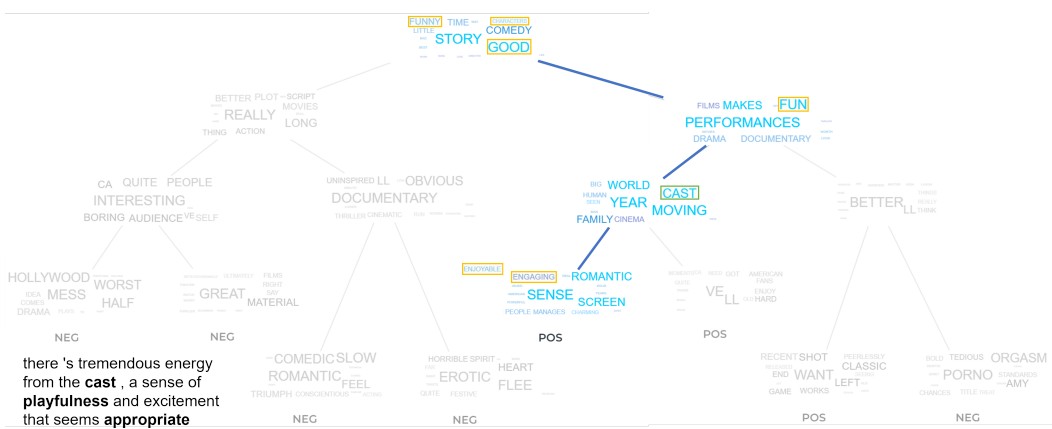

Figure 18: The local explanation decision tree generated for a correctly predicted **Positive** review in MR dataset based on fine-tuned RoBERTa embedding. Multiple positive comments like *good, fun, engaging, enjoyable* can be found in decision-making for the text describing something as *playful* and *appropriate*.

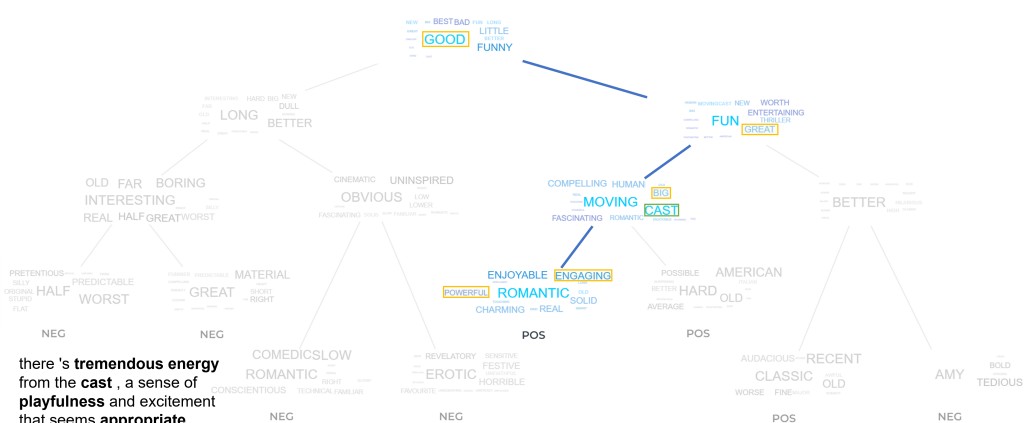

Figure 19: The local explanation decision tree generated for a correctly predicted **Positive** review in MR dataset based on fine-tuned RoBERTa embedding, with the POS filter to display only words which can be labelled as adjectives by the spaCy library.

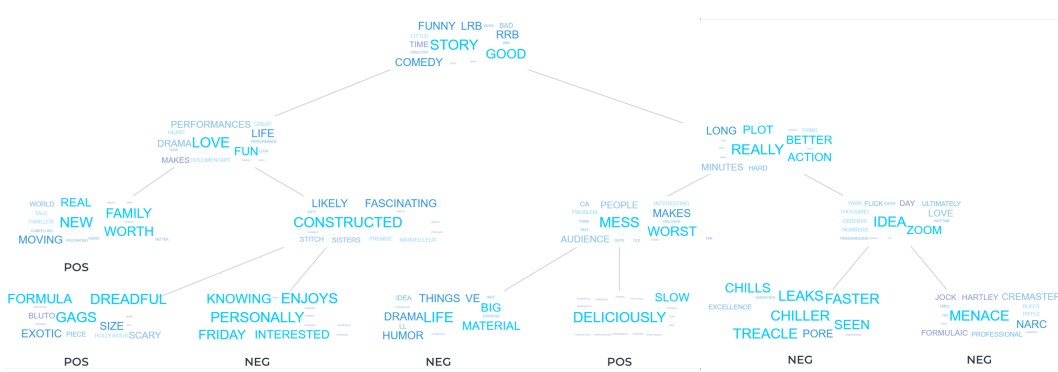

Figure 20: The global explanation decision tree generated on SST2 dataset based on fine-tuned RoBERTa embedding. Please take note that we explicitly limit the maximum depth of the tree during the training to prevent further branching out beyond 3rd children's level for better performance as well as interpretability, therefore the two children with the same class can be present. Words with negative connotations like *mess, worst* in human understanding tend to appear more in the prototype nodes or decision paths for the negative class, and similarly for the positive class with concepts like *worth, deliciously*.

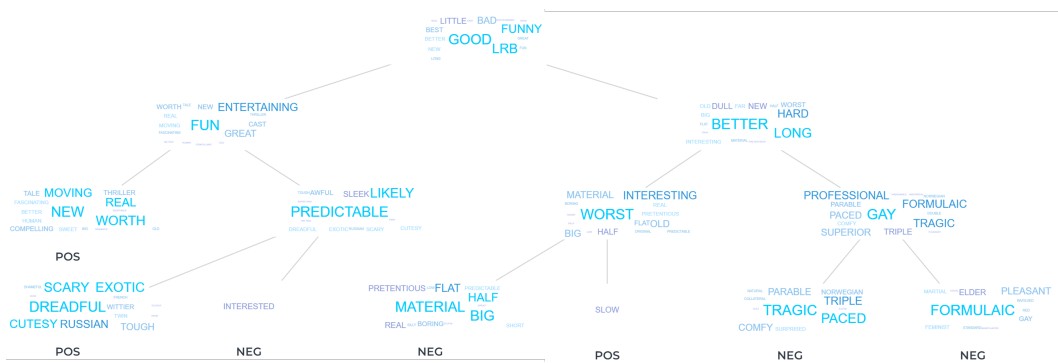

Figure 21: The global explanation decision tree generated on SST2 dataset based on fine-tuned RoBERTa embedding, with the POS filter to display only words which can be labelled as adjectives by spaCy library. More variations are of judgment other than common words like *good* or *bad* can be found when emphasising adjectives, such as *tragic, pretentious, dreadful* for the help of understanding the decision-making process. Errors inherited from the POS tagging model will lead to some non-adjectives remaining in the filtered global tree.

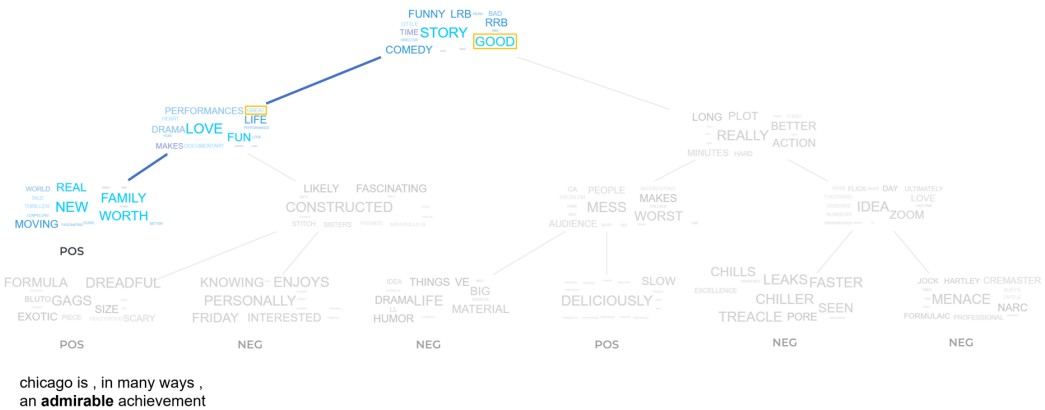

Figure 22: The local explanation decision tree generated for a correctly predicted **Positive** review in SST2 dataset based on fine-tuned RoBERTa embedding. Several positive concepts similar to *admirable* can be found in the decision path.

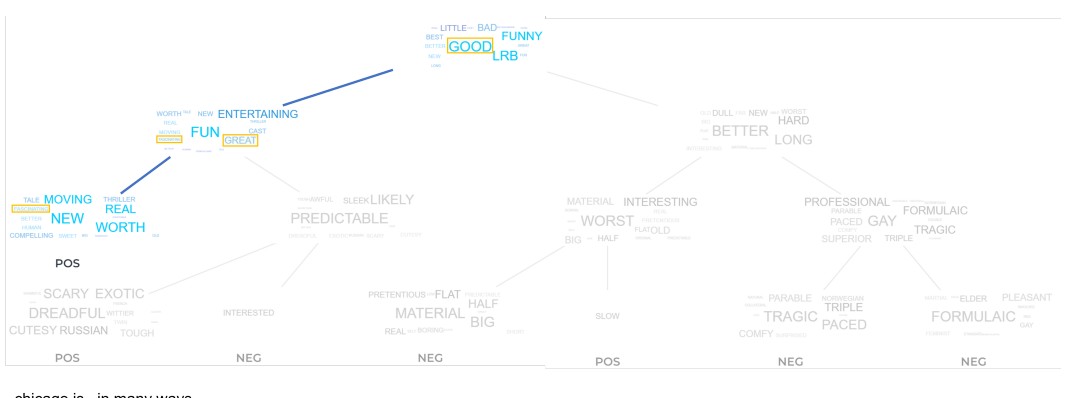

Figure 23: The local explanation decision tree generated for a correctly predicted **Positive** review in SST2 dataset based on fine-tuned RoBERTa embedding, with the POS filter to display only words which can be labelled as adjectives by spaCy library. Several positive concepts similar to *admirable* can be found in the decision path.

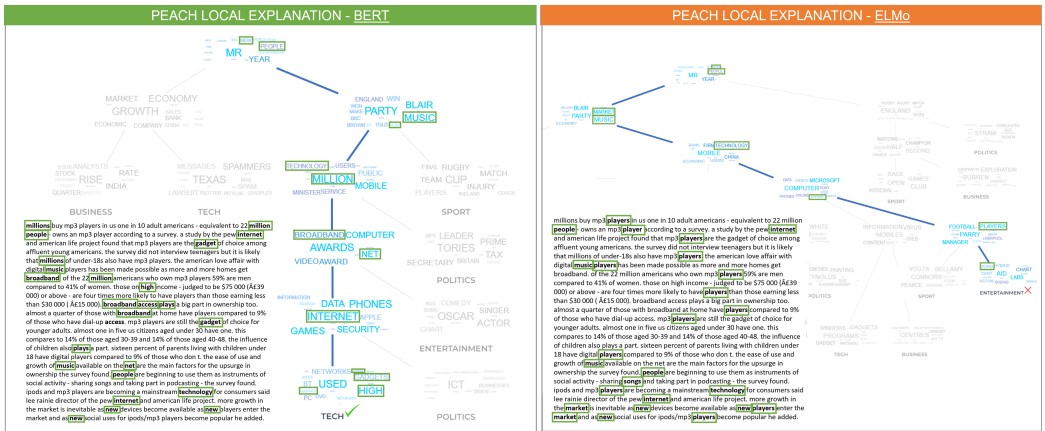

Figure 24: The local explanation decision trees generated for a **Tech** news article in BBCNews dataset based on fine-tuned BERT embedding compared to fine-tuned ELMo embedding, where PEACH(BERT) predicted correctly but PEACH(ELMo) predicted wrongly. While the BERT embedding fine-tuned by BBCNews has a clear decision-making path with exception nodes for technology classification, those with ELMo have the node that is lop-sided by the term 'players' and classified into either sports or entertainment.

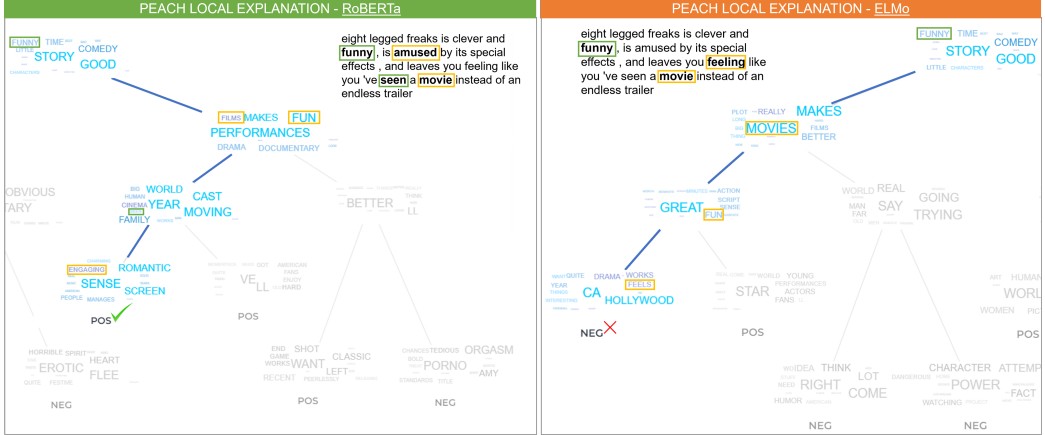

Figure 25: The local explanation decision trees generated for a **Positive** review in MR dataset based on fine-tuned RoBERTa embedding compared to fine-tuned ELMo embedding, where PEACH(RoBERTa) predicted correctly but PEACH(ELMo) predicted wrongly. While the successful embedding (RoBERTa) tends to have several adjectives that can highlight positive aspect (e.g. moving, engaging, romantic) or negative aspect (e.g. horrible, tedious), Elmo embedding does not understand the pattern of both with any remarkable adjectives patterns.

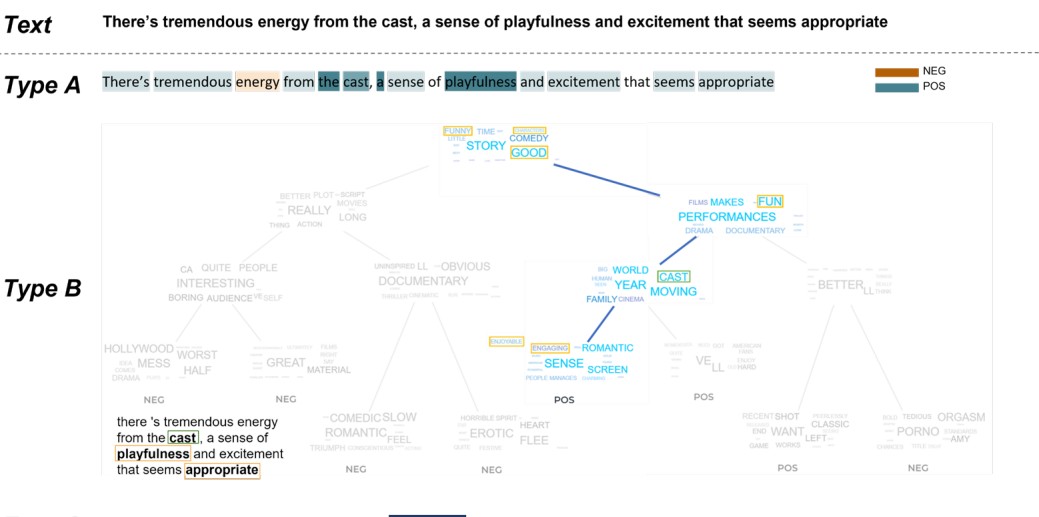

Figure 26: A sample case in the human evaluation to LIME and Anchor interpretation with our PEACH interpretation.

