# OpenReview forum: "PEACH: Pretrained-embedding Explanation Across Contextual and Hierarchical Structure"
_ICLR.cc/2024/Conference — ICLR 2024 Conference Withdrawn Submission_

### Official Review · Reviewer_1ymx · 2023-10-31

**Soundness:** 3 good
**Presentation:** 3 good
**Contribution:** 2 fair
**Rating:** 6
**Confidence:** 3

**Summary:**

The authors present “PEACH: Pretrained-Embedding Explanation across contextual and hierarchical structure” in which they propose an approach for finding a human-understandable interpretation of text classification with pre-trained language models.
The proposed approach works as follows: At first, PLMs are fine-tuned to the text classification tasks. The resulting vector representations of the CLS tokens are taken as Document embedding and are processed further. For this second feature processing step, the authors propose three variants, a Pearsons correlation based clustering, a k-means clustering based method, and a CNN. The goal of all three are to reduce the dimensions of the resulting feature matrix, and to identify the most informative dimensions for those features.
These features are then used to create a Decision Tree (based on the commonly used algorithms) that are built to solve the text classification task. Since the feature dimensions and the resulting decision tree nodes by themselves do not provide a human interpretable representation, the crucial task is to provide an interpretable decision tree node. This is accomplished by collecting all documents that pass through a specific node and use their tf-idf word-cloud representation as a visualization of this node. This procedure is done with all nodes, so that every node can be visually represented by a specific word cloud built from the documents that pass through them.
Their approach is then evaluated on a performance level, and on an interpretability level. In the performance evaluation, they compare their feature based decision tree approaches with the fine-tuned models. The interpretability is evaluated by human participants that evaluate how much they trust a given classification and a corresponding explanation.

**Strengths:**

-	The authors adopt an interpretability model from the computer vision domain to pre-trained language models and evaluate it with human study participants

-	The presentation quality makes the paper easy to follow

**Weaknesses:**

-	It is unclear to me how the improvements over the baselines are related to the interpretability aspect. It would be helpful to explain whether this is just a byproduct of the main research, or if it was a goal by itself.

-	Although the related work is listed and categorized, a little more explanation – especially for the LIME approach, which is compared to the proposed PEACH approach – would provide more background for the human evaluation task.

**Questions:**

-	Table 1: separate the results from Peach and the baseline models; maybe just add a horizontal line for easier comparison
-	Are the feature extraction methods crucial for the increased performance, or just happen to work well?

---

> ### Author Response · Authors · 2023-11-22
>
> Q1.
> * Thank you very much for the suggestion. We will add it to the camera-ready version.
>
> Q2.
> * It is very crucial, not only for performance but also for interpretability as well as training efficiency.
> * During our trial without any feature extraction methods, we noticed that only a small amount of dimensions of the embedding (out of 768 or 1024 dimensions) would be used as tree-splitting conditions, so we hypothesised that only a small amount of the embedding dimensions matter, probably due to similar concepts are learned across several dimensions, so we tried different feature extraction methods.
> * Without the feature extraction step,
>     - The trained decision tree tends to overfit. Our analysis for those trails shows that having too many features would lead to the generation of a bunch of exception rules for only one or two specific cases in the training set, which may not really apply to the testing set, making the performance a bit worse and the generated tree a lot more difficult to check and interpret (due to a large number of unnecessary rules.)
>     - The decision tree training is very slow and even impossible for large datasets, as most of the time is wasted calculating the information gain/gain ratio/gini coefficient for all of the input dimensions, even if most of them will not be used as splitting conditions.

---

### Official Review · Reviewer_HiZh · 2023-10-31

**Soundness:** 1 poor
**Presentation:** 2 fair
**Contribution:** 2 fair
**Rating:** 3
**Confidence:** 4

**Summary:**

- This paper introduces a tree-based explanation technique, titled PEACH (Pretrained-embedding Explanation Across Contextual and Hierarchical Structure), designed to elucidate the classification process of text-based documents through a tree-structured methodology.
- The authors demonstrate the utility of these explanations by employing word-cloud-based trees.
- Experimental results showcase that the classification outcomes achieved by PEACH either surpass or are on par with those derived from pretrained models.

**Strengths:**

This paper attempts to apply decision tree-based interpretations used in the CV field to the NLP domain. Theoretically, it is possible to combine certain explanatory elements under specific conditions to provide a more comprehensive, logical, and human-cognitively consistent interpretation.

**Weaknesses:**

This article asserts that the proposed word-cloud-based trees are human-interpretable and can clearly identify model mistakes and assist in dataset debugging. However, based on the examples illustrated in the figures of this paper (such as Figure 26), the word-cloud-based trees do not exhibit strong human-interpretability. They also do not align well with the semantics of the input sentences. From a human perspective, they do not provide clear explanations that would reduce cognitive load.

The experiments related to interpretability only compare the effectiveness of PEACH against LIME and Anchor, but do not conduct comparisons with the more human-intelligible Rationalization series of methods (Rationalizing Neural Predictions and subsequent series of studies).

There are several aspects of the paper that require improvement. The paper contains minor writing errors. The explanation of PEACH's interpretability in the introduction is not clear, the rationale behind word clouds is not analyzed, there is no systematic diagram, which hinders readability.

- In the description of MR in the DATASETS section, it mentions "two training documents," and it's unclear if this is an error.In the description of IMDB, there is a grammatical error; the word "and" should probably be changed to "has."
- In the IMDB column of Table 1, there are two bolded numbers, but it appears that there should be only one bolded number.
- In the article, the figures and tables are located far from their corresponding references, which hinders readability.
- The focus of PEACH should primarily be on interpretability, specifically on whether it can enhance human understanding of the text classification process. The article should emphasize this point in the introduction. However, the introduction lacks a direct comparison and analysis with previous methods, making it less intuitive. Figure 1 is placed in the introduction but is not explained, making it difficult to understand just by looking at it. In comparison, Figure 26 seems to be more fitting for inclusion in the introduction.
- The paper solely utilizes TF-IDF to construct word clouds without conducting an interpretability analysis. In the provided examples, it is challenging to discern the specific meanings of word clouds within the nodes, and there is limited overlap between the words in the word clouds and the sentences that need to be explained. This approach does not appear to offer stronger interpretability compared to previous methods.
- PEACH lacks a systematic diagram.

**Questions:**

In CV, decision tree nodes commonly employ specific image segments and representative patterns. In NLP, the choice of what to use as nodes is a topic of discussion. For the use of decision tree methods, employing word clouds as nodes in NLP doesn't seem as intuitive as using image segments as nodes in CV. At least, the examples provided in this article demonstrate this point.

---

> ### Author Response · Authors · 2023-11-22
>
> * We thank you very much for this valuable question and welcome any improvement on this part as future work.
> * Inspired by tree-based interpretable models in CV, we tried to adopt a tree-like visualisation that shows a clear reasoning pathway of trained PLM embeddings to a decision. With our PEACH, it is possible to explain the global trend of PLM representation and helpful for understanding the decision-making process of PLMs.
> * On top of this, we adopt the word cloud for visualising unstructured text data and getting insight into trends and patterns. Note that image segments are elements of an image, and, likewise, word tokens are elements of a textual document. Hence, we adopt this word token-based cloud for representing each node.

---

> > ### Comment · Reviewer_HiZh · 2023-11-22
> > **Thank you for the reply.**
> >
> > I will maintain the current rating.

---

### Official Review · Reviewer_TT7T · 2023-11-01

**Soundness:** 2 fair
**Presentation:** 2 fair
**Contribution:** 2 fair
**Rating:** 3
**Confidence:** 4

**Summary:**

The paper proposes a method for explaining the decisions made by a pretrained language model like BERT finetuned on a text classification dataset. The method consists of 4 steps: First, the pretrained model is finetuned in the typical way, i.e., by adding a classifier head on top of the CLS token. The CLS token representation at the last layer then constitutes the text embedding. Secondly, there is a feature dimensionality reduction step by either a) grouping individual dimensions by their correlation or b) by applying K-Means or c) by training a CNN on top. Third, a decision tree is trained on top of the reduced feature set. Finally, the nodes of the resulting decision tree are annotated with TFIDF statistics collected from the text examples associated with each node. These statistics are visualized as word clouds.
The method is evaluated on 5 text classification tasks with various datasets. The results indicate that the classification performance of the decision tree classifier reaches comparable performance to the finetuned model alone. A human evaluation is performed to compare the interpretability of the proposed methods to LIME and Anchor, two common baselines, which indicates that the proposed method performs drastically better. Additionally, some ablations and qualitative evaluations are performed.

**Strengths:**

* the paper proposes an intuitive and reasonable method
* the method is effective, both in the sense that the added interpretability doesn't come at the cost of classification performance and in the sense that the interpretability is much better than for the baseline methods according to human raters.
* the paper is relevant to the community

**Weaknesses:**

* It is not clear what problem the paper tries to solve. While it is motivated by the lack of trustworthyness of attention-score methods, there is no comparison in the end. Concrete research questions are missing. The introduction asks prominently "What should be considered a node of the decision tree in NLP tasks?", but this question is not revisited again.
* The datasets used during evaluation are quite easy. Many of them are just topic classification, which can be solved by extracting a few keywords. No advanced reasoning is required, which puts the value of the method in question.
* It is not clear how local explanations are obtained without manual inspection. No automatic procedure is described that would explain the annotations in the example figures.
* The choice of baselines for the interpretability evaluation is not motivated. Explanations of how the baselines work are missing. There is no explanation of why the proposed method works better by such a large margin, which means that the reader doesn't learn much from the paper.
* Individual components of the method's pipeline are well known techniques without any novelty apart from their straight forward combination.

**Questions:**

* visualizations are computed based on word statistics in the documents corresponding to a note. However, those are global statistics, so is there any guarantee that these can serve as local interpretations?
* since the visualizations are done based on TF-IDF, I wonder what is the added value of training the decision tree based on features from PLMs? What would be the performance if the decision trees were trained on word clusters directly?
* are the score differences in Figure 2 statistically significant? They seem quite small, and somewhat random.
* Section 4.3: Why did you choose LIME and Anchor as baselines? There is no description of how they work or how they were trained. The advantage of PEACH over them is very high, which remains unexplained without given further context of how these methods work different.
* How are the local explanations (Figure 3) generated? Some of the highlighted words have no exact representation in the word cloud. E.g., for ELMo, engaging is highlighted but not present in the word cloud. Conversely, entertaining is highlighted in the word cloud but not present in the text. Are these local explanations generated automatically?
* The low trustability of attention-score based interpretation methods are referenced as one of the main motivations in the introduction. Why are they not compared against as baselines?
* A study with human evaluators was conducted. Details regarding this study are missing, and it is not clear whether approval from an ethics board was seeked.

**Details Of Ethics Concerns:**

A study involving human evaluators was conducted in this paper. Details regarding the implementation of the study and the potential approval through an ethics board are missing.

---

> ### Author Response · Authors · 2023-11-22
>
> Q1.
> * Thanks for saying that it well-represents the global statistics. On top of this, we also demonstrate local interpretability by tracing the reasoning/inferencing process of each case with word statistics trends and highlighting the relevant keywords from each case related to the global word statistics trend. This would support human users in understanding the inference process of each case as a local interpretation. We believe this would be an important finding from the Natural Language Processing domain.
> * As can be seen in Figure 3, our PEACH can visualise the local interpretation that the finetuned ELMo representations do not capture the global pattern to distinguish between the positive class and negative reviews, leading to the wrong prediction for the positive review, and by checking the details of the word clouds against the review text itself we can identify the concept that the ELMo model gets confused about - ‘amusing/engaging’.
> * In the camera ready, we will add more case studies from some high-risk text classification datasets, which can enhance the importance of tracing the reasoning/inferencing process with word statistics for both global and local interpretation in the Natural Language Processing domain.
>
> Q2.
> * Thank you. We adopted a decision tree representation as a tool in order to interpret the pattern in the PLM representation. In NLP, many researchers/developers train/fine-tune PLM embeddings and use them for solving their downstream NLP tasks, such as text classification, question answering etc. However, there is no successful approach to convince users which PLM representation is trained well on a specific task, and which one can be used eventually. Due to this issue, users tend to try several PLM embedding methods and check the performance. Hence, we believe that it would be much better for users if there is any approach to explain how the trained PLM representations can understand/interpret the corpus (for the specific NLP task) that they trained.
> * The above aim is directly aligned with the original purpose of decision tree techniques, which aims to adopt tree-like visualisation in order to show a clear pathway to a decision. Hence, we use a decision tree visualisation by adopting word clusters based on word attention and word similarity in order to reason/infer the decision path for each NLP task.
>
> Q3.
> * As mentioned before, it is very important to train and select better PLM embeddings as an input feature for most NLP downstream tasks. Hence, we would like to share the indicators of how the embedding dimension can affect the performance.
> * We found that our PEACH clearly shows the performance pattern, especially when the corpus has larger classes and lower dimensions.
>
> Q4.
> * We chose LIME (2016) and Anchor (2018) as two well-known explainable AI methods, especially in the NLP domain. We referred to the original published paper, but we will update the detailed information on how these methods work in the camera-ready version.
>
>
> Q5.
> * Figure 3 is visualised for showcasing as the user, who uses our PEACH can easily check the decision path by inspection. The words highlighted in yellow (e.g. ‘engaging’ and ‘entertaining’ as mentioned by you) share similar meanings. Since words in word cloud visualisation try to demonstrate the representative concept for each node, it does not make sense to only look out for the exact match of the words between the document and the visualization (which is also not possible for datasets with diverse vocabulary).
>
> Q6.
> * LIME is the attention-score based method which we already use in the human evaluation study to compare our PEACH against with, in terms of interpretability.
> * LIME is just a framework to explain any classifier model based on the model weights (e.g. BERT, ALBERT, ELMo, etc., used as our baselines), and does not affect the original performance of the classifier model. That means the performance of LIME on BERT will be the same as the BERT performance itself, which we already included in Table 1.
>
> Q7.
> * We mentioned the brief human evaluation demographics and human evaluation interface in footnote 6 and figure 26. However, we will put more details about the human evaluation demographics, and approval will be attached to the appendix in the camera-ready version.